# Long-Term Synaptic Plasticity Tunes the Gain of Information Channels through the Cerebellum Granular Layer

**DOI:** 10.3390/biomedicines10123185

**Published:** 2022-12-08

**Authors:** Jonathan Mapelli, Giulia Maria Boiani, Egidio D’Angelo, Albertino Bigiani, Daniela Gandolfi

**Affiliations:** 1Department of Biomedical, Metabolic and Neural Sciences, Via Campi 287, University of Modena and Reggio Emilia, 41125 Modena, Italy; 2Centre for Neuroscience and Neurotechnology, University of Modena and Reggio Emilia, 41125 Modena, Italy; 3Department of Brain and Behavioral Sciences, Neurophysiology Unit, Via Forlanini 6, 27100 Pavia, Italy; 4Brain Connectivity Center (BCC), IRCCS C. Mondino, Via Mondino 2, 27100 Pavia, Italy

**Keywords:** gain modulation, long-term plasticity, cerebellum, granule cell, computational modeling

## Abstract

A central hypothesis on brain functioning is that long-term potentiation (LTP) and depression (LTD) regulate the signals transfer function by modifying the efficacy of synaptic transmission. In the cerebellum, granule cells have been shown to control the gain of signals transmitted through the mossy fiber pathway by exploiting synaptic inhibition in the glomeruli. However, the way LTP and LTD control signal transformation at the single-cell level in the space, time and frequency domains remains unclear. Here, the impact of LTP and LTD on incoming activity patterns was analyzed by combining patch-clamp recordings in acute cerebellar slices and mathematical modeling. LTP reduced the delay, increased the gain and broadened the frequency bandwidth of mossy fiber burst transmission, while LTD caused opposite changes. These properties, by exploiting NMDA subthreshold integration, emerged from microscopic changes in spike generation in individual granule cells such that LTP anticipated the emission of spikes and increased their number and precision, while LTD sorted the opposite effects. Thus, akin with the expansion recoding process theoretically attributed to the cerebellum granular layer, LTP and LTD could implement selective filtering lines channeling information toward the molecular and Purkinje cell layers for further processing.

## 1. Introduction

The signal processing in neuronal circuits requires the dynamic regulation of synaptic responses. The sensitivity to input transformation is in turn modulated by integrating a wide series of mechanisms such as background noise integration [1,2], short-term synaptic plasticity [3] or shunting inhibition [4]. Moreover, additional mechanisms persistently altering the balance between synaptic inputs can actively regulate the neuronal input–output relationship. In this scenario, long-term synaptic plasticity [5,6,7,8] has been shown to modulate neuronal sensitivity to input transformation, leading to alterations in the input–output transfer function. Persistent changes in synaptic efficiency induced by LTP and LTD are expected to change transmission parameters such as spike-timing [9] or frequency range dependency [10]. LTP and LTD are thought to provide an elementary mechanism for learning and memory [11,12]. In general, a change in synaptic weight is expected to modify the activation of circuit elements implementing complex functions [13,14]. Moreover, self-organization and other computational effects are expected to emerge from the topological organization of excitatory and inhibitory connections (e.g., Kohonen layer, Hopfield recurrent networks, Sejinowsky Boolean operators [15,16,17]). However, these theoretical models assume that LTP and LTD are just “synaptic weight” changes and omit any details on the way LTP and LTD could dynamically control spike processing in local circuits [18]. In principle, LTP and LTD should be able to affect the timing, number and frequency of spikes emitted in response to incoming spike bursts. Even more, by modifying mechanisms sensitive to the incoming burst patterns (such as the NMDA current or neurotransmitter release dynamics), LTP and LTD could regulate the gain and frequency dependence of signal transmission.

In the cerebellar granular layer, a brain region where communication in the time domain is particularly important, the spatiotemporal transformation of incoming signals is believed to occur mostly through gain regulation implemented by mechanisms involving GABA and NMDA receptors [19]. In particular, factors that have been proposed to regulate mossy-fiber granule cell gain are the levels of shunting GABAergic inhibition [3,20] and of NMDA current activation in the glomerulus [21,22]. GABA and NMDA receptors can indeed differentially regulate the frequency dependence of burst transmission between the granular and molecular layer [23] and the recombination of spatial patterns [19]. The number and pattern of spikes emitted by GrCs depend on numerous factors, including the intensity of local synaptic inhibition and the consequent level of activation of NMDA receptors, which markedly enhance burst transmission [21]. Moreover, the level of activation of NMDA receptors during repeated mossy fiber bursting is critical to inducing LTP and LTD at the MF-GRC synapse [24,25]. Mf-GrC long-term plasticity is induced through a Ca2+-dependent process implementing a BCM-like relationship [25]. LTP and LTD are expressed through changes in neurotransmitter release altering the response dynamics of both AMPA and NMDA receptors [26]. Furthermore, in the glomerular structure, other plastic mechanisms involving Golgi cells have been shown to contribute regulating signal transmission through the granular layer [7,27]. Thus, LTP and LTD are potentially capable of implementing the complex regulation of the transmission function through the granular layer in a way that is not simply explained by a “weight change”. Since the temporal dynamics of neurotransmission are also changed, LTP and LTD remarkably modulate the time of emission of granule cell spikes [28,29]. Furthermore, since NMDA receptor activation is highly non-linear [30], the NMDA receptors cause a frequency-dependent regulation of neurotransmission gain through the granular layer [19,31]. Finally, the spatial organization of afferent synapses and local inhibitory circuits is rather complex [23,32]. The presence of lateral inhibition causes a center-surround organization of activity reflecting local changes in the excitatory/inhibitory balance [33]. It is therefore conceivable that LTP and LTD can markedly reconfigure the spatiotemporal organization of burst transmission through the granular layer. Nevertheless, although it has been recently demonstrated that long-term plasticity can influence the spatiotemporal organization of the signal transmitted through the granular layer [5,10], clear evidence that LTP and LTD affect gain transmission at the single-cell level is still missing.

Here, we show, by using a combination of patch-clamp intracellular recordings and mathematical modeling, that long-term plasticity at the cerebellar input stage can alter the gain bandwidth of mossy fiber burst transmission. These effects depended on subtle changes in spike generation such that the LTP at the mf-GrC-generated spikes had a smaller delay, higher probability, and higher temporal precision than the LTD. This mechanism could implement selective transmission lines channeling information through the cerebellum input stage, bearing implications for further spike processing in the molecular and Purkinje cell layers which are supposed to encode information through spike rate modulation [34]. The relationship between this finding and the expansion recoding process attributed to the granular layer—for example, in the Adaptive Filter Model theory [35]—is discussed.

## 2. Materials and Methods

Experiments were performed using Sprague–Dawley rats at postnatal day P17–P24 (internal breeding, Charles-Rivers (Calco, Lecco, Italy)). All experiments were conducted in accordance with international guidelines from the European Community Council Directive 86/609/EEC on the ethical use of animals and were approved by the Ethical committee of the Italian Ministry of Health and by the Ethical Committee of the University of Modena and Reggio Emilia. Furthermore, the study was carried out in compliance with the ARRIVE guidelines (http://www.nc3rs.org.uk/page.asp?id=1357 (accessed on 30 June 2020). Animals (*n* = 28) were chosen independently of gender, and a total number of 50 cells were employed to perform this research.

### 2.1. Recordings in Acute Cerebellar Slices

Patch-clamp recordings were obtained from parasagittal cerebellar slices obtained as described previously [36,37]. Briefly, rats were deeply anesthetized with isoflurane (Sigma-Aldrich, St. Louis, MO, USA) and decapitated. The cerebellum was removed, and the vermis was isolated and fixed on a vibroslicer stage (VT1000S, Leica Microsystems, Nussloch, Germany) with cyanoacrylic glue. Acute 200 µm-thick slices were cut in a cold cutting solution containing (in mM): 130 K-gluconate, 15 KCl, 0.2 EGTA, 20 HEPES and 10 glucose; pH adjusted at 7.4 with NaOH. Slices were incubated at 32 °C for at least 1 h before recordings in an oxygenated extracellular Krebs solution containing (in mM): 120 NaCl, 2 KCl, 1.2 MgSO_4_, 26 NaHCO_3_, 1.2 KH_2_PO_4_, 2 CaCl_2_ and 11 glucose (pH 7.4 when equilibrated with 95% O_2_ and 5% CO_2_). Slices were then transferred to a recording chamber on the stage of an upright microscope (Zeiss Axioexaminer A1, Oberkochen, Germany) and perfused at 1.5 mL min^−1^ with an oxygenated Krebs solution maintained at 32 °C with a thermostatic controller (Multichannel system, Gmbh, Reuntlingen, Germany). Slices were immobilized with a nylon mesh attached to a platinum Ω-wire.

Whole-cell current-clamp recordings were made in the whole-cell patch-clamp configuration from granule cells, as reported previously [38,39]. Cells were preferentially chosen in the central lobules of the vermis (from IV to VII), and recordings were obtained using an Axopatch 200B amplifier (Molecular Devices, Union City, CA, USA) (−3 dB; cut-off frequency = 2 kHz). Recordings were digitized at 20 kHz using pClamp 9 (Molecular Devices) and a Digidata 1322A A/D converter (Molecular Devices). Patch pipettes were made with a vertical puller (model PP-830, Narishige, Tokyo, Japan) from borosilicate glass capillaries and filled with the following solution (in mM): 126 K-gluconate, 8 NaCl, 15 glucose, 5 HEPES, 1 MgSO_4_, 0.1 BAPTA-free, 0.05 BAPTA-Ca^2+^, 3 ATP, 100 µM GTP; pH adjusted to 7.2 with KOH. This solution maintained resting free-[Ca^2+^] at 100 nM, and the pipettes had a resistance of 7–10 MΩ before seal formation. The stability of patch-clamp recordings can be influenced by modifications of series resistance and neurotransmitter release. To ensure that the series resistance remained stable during the recordings, passive cellular parameters were extracted in a voltage-clamp by analyzing the current relaxation induced by a 10 mV step from a holding potential of −70 mV. According to previous reports [21,40,41], the transients were reliably fitted with a monoexponential function yielding a membrane capacitance (Cm) of 2.4 ± 0.3 pF (*n* = 18), a membrane resistance (Rm) of 2.3 ± 0.4 GΩ (*n* = 18) and a series resistance (Rs) of 17.5 ± 1.3 MΩ (*n* = 18). Mossy fibers were stimulated with a bipolar tungsten electrode (Clark Instruments, Pangbourne, UK) via a stimulus isolation unit at a stimulation intensity of ±4–12 V (duration 100 μs) from a membrane potential between −70 and −60 mV (mean 65.2 ± 3.7, *n* = 16). From a comparison with previous data [21,40,42], between one and three mossy fibers were stimulated per granule cell.

In order to investigate the frequency-dependent properties of the granular layer response, stimuli were repeated as in [10,23] at frequencies between 10 and 500 Hz in random order to prevent systematic effects due to adaptation processes. Each trial was composed of five pulses delivered to the mossy fiber bundle and was repeated 10 times at a frequency of 0.1 Hz to account for response variability.

### 2.2. Long-Term Plasticity Induction and Analysis

Granule cells responses were monitored in voltage-clamp configuration at the beginning and at the end of the recordings. The induction of LTP was performed by delivering a TBS (Theta-Burst Stimulation) protocol in current clamp configuration from a holding potential of −60 mV (10 pulses at 100 Hz delivered eight times every 250 ms). The analysis of changes induced by long-term plasticity was performed as in [24,43]. Briefly, granule cells were held at a holding potential of −70 mV, and EPSCs were evoked with a 50 Hz paired pulse stimulation. Then, 50 consecutives traces were evoked, and EPCSs amplitudes were averaged, yielding 29.7 ± 4.4 pA (*n* = 18) [42]. The paired pulse ratio (PPR) was taken as the ratio between the first and the second evoked EPSC (20 ms interval) peak, and the average value was 0.78 ± 0.05 (*n* = 18) [42]. Moreover, EPSPs evoked by a single stimulus were used to assess the record stability and the variations induced by TBS. On average, the EPSPs amplitude was 11.4 ± 0.9 mV (*n* = 18 cells).

### 2.3. Gain Function Analysis

Gain functions were obtained as in [23] by analyzing the input–output variations in the number of spikes, the first-spike standard deviation, the spike probability (as the probability of emitting at least one spike in the burst) and the average maximum depolarization (sc, fssd, sp and amd respectively). At all tested frequencies, sc, fssd, sp and amd were measured, normalized between the extreme values for each cell and rescaled between 0 and 1. Thus the gain value approaches one when the number of emitted spikes, the membrane average depolarization and the spike probability are at the maximum and the first spike standard deviation is at the minimum. Then, the four parameters (sc, fssd, sp, amd) were summed, yielding a compound gain function ranging from 0 to 4 and representing the gain modulation in the given cell.

Gain curves *g(f)* were fitted with a sigmoidal-shaped function of input frequency (*f*) of the form:(1)$$g\left(f\right)=\frac{\left(A_{1}-A_{2}\right)}{\left(1+{\left(\frac{f}{f_{c}}\right)}^{p}\right)}+A_{2}\;$$
where *A*_1_ and *A*_2_ are the initial and final amplitude, *f_c_* is the cut-off frequency and *p* is the power of the function (MATLAB). Data are reported as the mean ± SEM and compared for their statistical significance by a paired Student’s *t*-test, unless stated otherwise (a difference was considered significant at *p* < 0.05).

### 2.4. Mathematical Modeling

A realistic multi-compartmental model was used to perform simulations. The model is written in NEURON (v 8.9) and Python (v.3.8.8), and simulations were run on a personal computer AMD Rizen 5 3600 6-Core Processor 3.60 GHz. The synaptic models of single neurons were adapted from the original scheme reported in [28] and further refined in [44]. The model could reproduce the kinetics and size of the postsynaptic currents during repetitive synaptic transmission at the different synapses. These models accounted for vesicular dynamics, neurotransmitter spillover and receptor gating (including multiple closed, desensitized, and open states) but not for quantal release mechanisms. The dynamics of synaptic responses were fully determined by the kinetic constants of synaptic and neuronal models. Axonal conduction times were considered negligible, and the transmission delay was set to 1 ms for all the synapses.

Briefly, the GrC model was adapted from [28] by applying appropriate Q10 corrections. In addition, the GABA leakage conductance was increased by two times (60 µS/cm^2^), the inward rectifier K+ conductance was increased by 1.5 times (1350 µS/cm^2^) and the leakage reversal potential was adjusted to restore the resting potential to −70 mV. With this asset, the GrC model properly reproduced the responses to the current injection at 37 °C (data not shown) and the spike trains observed in vivo [45].

The multicompartmental model implemented the morphological structure, channel localization and dynamics. The soma had a 5.8 μm diameter attached to four identical unbranched dendrites (15 μm diameter) subdivided into 4 compartments and to the axon (0.75 μm diameter) with 5 proximal and 30 distal compartments. For each compartment, *V* was obtained as the time integral of the equation:(2)$$\frac{dV}{dt}=\frac{1}{\tau_{m}}\left(V-\frac{\sum_{i}g_{i}\left(V-V_{i}\right)+\sum_{syn}g_{syn}\left(V-V_{syn}\right)+\sum_{i}g_{br}\left(V-V_{br}\right)}{g_{tot}}\right)$$
where *τ_m_* = *R_m_C_m_* is the membrane time constant (*R_m_* = 1/*g_tot_* and *C_m_* are the membrane resistance and capacitance); *g_i_*, *g_syn_* and *g_br_* are the conductances through voltage-dependent channels, synaptic channels, and neighboring branches; *V_i_* and *V_syn_* are the reversal potentials for voltage-dependent and synaptic channels, while *V_br_* is the membrane potential of neighboring branches.

Intracellular calcium concentration [Ca^2+^] dynamics were calculated with the equation:(3)$$\frac{d\left[{\mathrm{Ca}}^{2+}\right]}{\mathrm{dt}}=-\frac{I_{{\mathrm{Ca}}^{2+}}}{2F\cdot A\cdot d}-\left({\;\beta }_{{\mathrm{Ca}}^{2+}}\left(\left[{\mathrm{Ca}}^{2+}\right]-{\left[{\mathrm{Ca}}^{2+}\right]}_{0}\right)\right)$$
where d is the depth of a shell adjacent to the cell surface of area *A*, Ca^2+^ determines the loss of calcium ions from the shell approximating the effect of fluxes, ionic pumps, diffusion, and buffers, [Ca^2+^]_0_ is resting [Ca^2+^] and F is the Faraday’s constant. The mathematical representations of g_i_ and g_syn_ (Equation (2)), which can be voltage- and calcium-dependent, are fully reported in previous papers [28,46]. Channel localization and parameters are reported in Table 1 (adapted from [44]). Sodium currents accounted for the three components reported in granule cells, namely, Na_t_, Na_p_ and Na_r_ (transient, persistent, and resurgent; Table 1). The Potassium delayed rectifier (K_DR_) channels were placed together with Na channels and accounted for spike repolarization. In GrCs, intracellular Ca^2+^ transients depend on voltage-dependent calcium channel activation in dendrites. Ca^2+^ and K_Ca_ currents were placed in the dendritic endings with intracellular Ca^2+^ dynamics. The parameters adopted in Equation (3) were d = 200 nm, Ca = 0.6 and [Ca]_0_ = 100 nM. K_A_ channels were placed in the soma together with K_IR_ and K_SLOW_. The model included a leakage current (Lkg1) and a tonic inhibition component (Lkg2; GABA_A_ receptors activity).

#### Synaptic Dynamics

The synaptic model of GrCs was implemented with four identical independent synapses with pre- and postsynaptic independent dynamics.

Mossy fiber-granule cell EPSCs short-term dynamics have been derived from the Tsodyks and Markram three-state scheme [47], with X representing the transmitter resources available for release, Y representing the released transmitter and Z representing the recovered transmitter. The transition between states is governed by first-order reactions according to a first-order differential equation system:$$\left\{\begin{array}{c} \frac{dX}{dt}=\frac{Z}{\tau_{R}}-P\cdot X\cdot \delta \left(t-t_{spike}\right)\\ \frac{dY}{dt}=-\frac{Y}{\tau_{t}}+P\cdot X\cdot \delta \left(t-t_{spike}\right)\\ \frac{dZ}{dt}=\frac{Y}{\tau_{I}}-\frac{Z}{\tau_{R}}\\ \frac{dP}{dt}=-\frac{P}{\tau_{F}}+p\left(1-p\right)\cdot \delta \left(t-t_{spike}\right) \end{array}\right.$$
with *τ_R_* being the recovery time constant of the releasable transmitter, *τ_F_* being the time constant of facilitation, *τ_I_* being the time constant of inactivation, *P* being the release probability, *p* being its initial value and *δ* being Dirac’s delta function. The system is solved analytically by integrating differential equations in intervals $\left[t_{spike,n},\;\;t_{spike,n+1}\right]$ for initial conditions at time $t_{spike,n}$. At times *t* = *t_spike_*, the values of functions are changed abruptly by *pX* or *P*(1−*p*). Consequently, when a spike arrives, a proportion *P* of the resource *X* is transferred to *Y*. The depletion of the resource *X* causes synaptic depression (another component depends on postsynaptic receptor desensitization). Synaptic facilitation is governed by *p* activity-dependence. Granule cell postsynaptic responses are generated both through direct release and glutamate spillover. For AMPA receptors, glutamate concentration was obtained by combining a synaptic pulse (*T_s_*) with a diffusion wave (*T_d_*). The effective glutamate concentration in the cleft is therefore:$$T=T_{s}+T_{d}=Y\times \left[T_{s}\left(max\right)+T_{d}\left(max\right)\right]$$
where *T_s_(max)* and *T_d_(max)* are maximum concentrations. NMDA receptors, which are largely extra-synaptic, were activated by glutamate diffusion, *T_d_*. Glutamate binding activates first-order transitions kinetic schemes, leading to the open state, *O(T)*.
$$I=g\cdot \Delta V=gmax\cdot O\left(T\right)\cdot \left(V-V_{rev}\right)$$
where *V_rev_* is the ionic reversal potential, and *g_max_* is the maximum synaptic conductance for either NMDA or AMPA channels.

AMPA receptors were modeled as a simple three-state AMPA receptor kinetic scheme with k_o+_ = 5.4 ms^−1^, k_o__ = 0.82 ms^−1^, k_d+_ = 1.12 ms^−1^, k_d__ = 0.013 ms^−1^, K_B_ = 0.44 mM and S = T_2_/(T + KB)^2^. NMDA kinetic schemes were modeled with two identical closed states, and the desensitized state is entered from the second closed state, with k_1+_ = 5 mM^−1^ms^−1^, k_1__ = 0.1 ms^−1^, k_2+_ = 5 mM^−1^ms^−1^, k_2__ = 0.1 ms^−1^, k_o+_ = 0.03 ms^−1^, k_o__ = 0.966 ms^−1^, k_d+_ = 0.00012 ms^−1^ and k_d__ = 0.009 ms^−1^.

The Golgi cell model [48,49] consisted of five compartments allowing for a minimal description of the Golgi cell electrotonic structure. All voltage- and Ca^2+^-dependent mechanisms (see below) were placed in the somatic compartment. The model included 12 voltage-dependent conductances placed into the soma. Only the Ca^2+^ Nernst potential was updated during simulation.

Voltage was obtained as the time integral of the equation
$$\frac{dV}{dt}=-\frac{1}{C_{m}}\cdot \left\{\sum \left[g_{i}\cdot \left(V-V_{i}\right)\right]+i_{inj}\right\}$$
where *V* is the membrane potential, *C_m_* is the membrane capacitance, *g_i_* are ionic conductances, *V_i_* are reversal potentials (the subscript i indicates different channels) and *I_inj_* is the injected current. *I_AHP_* was simulated using a Markov gating scheme, while all other membrane conductances were represented using Hodgkin–Huxley-like models of the type
$$g_{i}=G_{max_{i}}\cdot x_{i}^{zi}\cdot y_{i}$$
where *G_maxi_* is the maximum ionic conductance, *x_i_* and *y_i_* are state variables (probabilities ranging from 0 to 1) for a gating particle and *z_i_* is the number of such gating particles in ionic channel *i*. *x* and *y* (the suffix *i* is omitted) were related to the first-order rate constants a and b by the equations
$$x_{\infty }=\frac{\alpha_{x}}{\alpha_{x}+\beta_{x}},\;\;\;\;y_{\infty }=\frac{\alpha_{y}}{\alpha_{y}+\beta_{y}}\;\;$$
$$\tau_{x}=\frac{1}{\alpha_{x}+\beta_{x}},\;\;\;\;\tau_{y}=\frac{1}{\alpha_{y}+\beta_{y}}$$
where a and b are functions of voltage.

The state variable kinetics were
$$\frac{dx}{dt}=\frac{x_{\infty }-x}{\tau_{x}},\;\;\;\;\frac{dy}{dt}=\frac{y_{\infty }-y}{\tau_{y}}\;\;\;\;$$

The intracellular Ca^2+^ concentration, [Ca^2+^], was calculated through the equation
$$\frac{d\left[{\mathrm{Ca}}^{2+}\right]}{dt}=-\frac{I_{{\mathrm{Ca}}^{2+}}}{2F\cdot A\cdot d}-\left(\;\beta_{{\mathrm{Ca}}^{2+}}\left(\left[{\mathrm{Ca}}^{2+}\right]-{\left[{\mathrm{Ca}}^{2+}\right]}_{0}\right)\right)$$
where d is the depth of a shell adjacent to the cell surface of area *A*, $\beta_{{\mathrm{Ca}}^{2+}}$ determines the loss of Ca^2+^ ions from the shell approximating the effect of fluxes, ionic pumps, diffusion and buffers and [Ca^2+^]_0_ is the resting [Ca^2+^].

In the GoC model, the capacitances of soma, dendrites and axon were set to 23, 32 and 90 pF, respectively, summing up to a total cell capacitance of 145 pF. The specific axial resistance of axon and dendrites was set to 100 Ωcm, the specific membrane resistance was set to 47.6 KΩ cm^2^ and the specific membrane capacitance was set at 1 mF/cm^2^. Morphologies were generated to comprise a spherical soma (diameter = 27 μm), a single axonal compartment (1200 μm-long and 2.4 μm-thick) and three dendritic compartments (each 113 μm-long and 3 μm-thick).

In order to conform to in vivo conditions, all models were adapted from their original temperature Torig to Tsim = 37 °C using the correction factor Q10 = (Tsim − Torig)/10. We have used: Q10 = 3 for ionic channel gating, Q10 = 2.4 for receptor gating, Q10 = 1.5 for ionic channel permeation, Q10 = 1.3 for neurotransmitter diffusion, Q10 = 3 for Ca^2+^ pumps and buffers and Q10 = 1.3 (GrC) or 1.7 (GrC) for intracellular Ca^2+^ diffusion. Following adaptation at 37 °C, the models were matching with recordings at this same temperature (data not shown).

The GoC model was adapted from [48,49] by applying appropriate Q10 corrections. Without needing any further change, the GoC model properly reproduced responses to the peripheral stimulation observed in vivo [50].

All the results shown in the manuscript have been obtained by setting the temperature in the simulation at 30°, which accounts for a thermal dispersion of the solution in the recording chamber (from 32° at the border where the perfusing syringe is located to about 30° in the middle of the chamber).

The glutamatergic mf-GrC synapses take part in the formation of the cerebellar glomerulus and activate AMPA and NMDA receptors. Using a probability of release of 0.42, the model was able to faithfully reproduce postsynaptic currents recorded at 37 °C in vitro [28] and in vivo [45]. The short-term dynamics of the mf-GrC synapse were modeled as in [10,28], adopting a time constant of the recovery from depression τREC = 8 ms, which was derived from in vivo measurements [51] and allowed to reproduce the natural dynamics of short-term plasticity (the time constants of presynaptic facilitation and vesicle inactivation were set to τ_facil_ = 5 ms and τ_I_ = 1 ms, respectively).

The mf-GoC synapses are similar in several aspects compared to the mf-GrC synapses. They are also located within the cerebellar glomerulus and are glutamatergic, activating both AMPA and NMDA receptors [32]. The mf-GoC synapse was adapted from the mf-GrC synapse model (see above) to reproduce a peak of postsynaptic current of −66 pA. The release probability and vesicle cycling parameters were set at the same values as those at the mf-GrC synapse.

The GrC-GoC synapses are formed by PFs onto GoC apical dendrites in the molecular layer. These glutamatergic synapses activate AMPA, NMDA and kainate receptors (See Appendix A). During repetitive stimulation, the AMPA current shows synaptic depression, while the kainate and NMDA currents show slow temporal summation. AMPA and NMDA currents were taken from the MF-GrC synapses, and the kainate receptor current was modified from the AMPA kinetic scheme. The release probability was 0.1, and the vesicle cycling parameters were set at the same values as those at the MF-GrC synapse. The AA contacts GoC basolateral dendrites in the granular layer; these synapses activate AMPA and NMDA only; their maximal conductance was estimated to be ~2 times higher than AMPA and NMDA currents of PF-GoC synapses. Additionally, in this case, AMPA and NMDA currents were taken from the MF-GrC synapse; the release probability and vesicle cycling were also set at the same values.

The GoC-GrC synapses are GABAergic and impinge on GrC dendrites within the glomerulus. The GABA-A receptor schemes comprised channels with fast (α1) and slow (α6) kinetics and GABA spillover generating the transient and sustained components of inhibition observed experimentally. The GoC-GrC model has been derived from [52], which in turn has been tuned on the experimental data shown in [53]. The parameters describing presynaptic dynamics were: release probability = 0.35, τREC = 36 ms, τfacil = 58.5 ms and τI = 0.1 ms, respectively [52].

During repetitive stimulation, the GrCs membrane potential was set at an initial value of −65 mV through a constant current injection, whereas the expression of long-term plasticity was monitored in a voltage configuration holding GrCs at -70 mV.

Granule cell activity was simulated in response to variable configurations of stimulated mfs and GoCs. The microcircuit is reported in Appendix A. The model was further modified to better reproduce experimental results in terms of excitability and inter-trial variability. The number of release sites (nrel) was increased from 4 to 6, and the max conductance (gmax) of AMPA and NMDA channels at the MF-GrC synapse was equally divided between the release sites: each site was equipped with gmax = 1200/nrel for AMPA and gmax = 18800/nrel for NMDA. The mf-GrC synapse was equipped with a stochastic model [31] to better mimic the variability of the release mechanism experimentally observed.

The synaptic plasticity was modeled as a purely presynaptic mechanism by increasing or decreasing the release probability of 50% of the initial value (mf-GrC LTP 0.63, LTD 0.21; GoC-GrC LTP 0.525, LTD 0.175) [28].

## 3. Results

In this paper, we have used intracellular patch-clamp recordings in acute rat cerebellar slices to evaluate changes caused by long-term plasticity to the transmission properties of the mf-GrC synapse. The simulations of gain changes induced by long-term plasticity allowing for the assessment of the impact of LTP and LTD on filtering properties have been performed by assembling, in a simplified version of the cerebellar input stage, i) a multicompartmental model of the GrC (see Methods and [10,28,36] for a detailed description of the model), ii) a multicompartmental model of the GoC (see Methods and [10,36,48,49] for a detailed description) and iii) biologically realistic synapses endowed with short-term dynamics (see Methods and [28,36,52] for a detailed description). This simple configuration was adopted to implement a bottom-up [54] exploration of the cellular and molecular mechanisms underlying gain modulation.

### 3.1. The Effect of Long-Term Synaptic Plasticity on a Single Granule Cell

We have recently shown [10], by using a combination of two-photon recordings [55,56], VSD imaging [57,58] and computational modeling [59], that long-term plasticity at the mf-GrC synapse induces a signal reconfiguration in the cerebellar granular layer. To investigate the origin of such changes, intracellular patch-clamp recordings were performed in acute cerebellar slices, and Long-Term Plasticity was induced by delivering Theta Burst Stimulation (TBS; Materials and Methods) to mossy fiber-granule cell synapses (Figure 1A). It has been shown that mf-GrCs can differentially undergo LTP or LTD depending on the GrCs depolarization attained during the induction [60]. The effects of TBS were evaluated by measuring changes in both Excitatory Post-Synaptic Potentials (EPSPs) and Excitatory Post-Synaptic Currents (EPSCs) (Figure 1B,C). In particular, paired pulsed stimuli (50 Hz) in the voltage clamp configuration (GrCs at −70 mV) were used to evaluate the presence of presynaptic expression mechanisms. A persistent increase in the evoked responses was detected in 37.5% of the recorded cells (6 out of 16), while the remaining showed a persistent decrease (62.5%). In GrCs showing LTP, EPSCs increased by 28.5% ± 3.1 (*n* = 6; *p* < 0.001; Figure 1B) and EPSPs increased by 31.2% ± 2.8 (*n* = 6; *p* < 0.01; data not shown), whereas PPR decreased by −27.4% ± 4.4 (*n* = 6; *p* < 0.001; Figure 1B). Accordingly, in cells showing LTD EPSCs, the amplitude significantly decreased (−36.8% ± 8.1; *n* = 10; *p* < 0.001; Figure 1C) together with EPSPs (−27.8% ± 2.9; *n* = 10; *p* < 0.01; Figure 1D), whereas the PPR increased (23.8% ± 10.6; *n* = 10; *p* < 0.01; Figure 1C). These results confirm that long-term plasticity did not differ from previous reports obtained at this same synapse [28,42,43].

### 3.2. The Impact of LTP and LTD on Filtering Properties

In the same experiments, repetitive stimulation at different frequencies was adopted to assess gain-frequency dependencies (Figure 2A). Neuronal gain modulation was hence evaluated as the input–output relationship of the firing frequencies [2] for cells responding with repetitive firing (Figure 2A,B). The resulting curves showed sigmoidal shapes with cutoff frequencies between 20 and 100 Hz (Figure 2C, average cutoff (78 ± 15 Hz; *n* = 16 cells Figure 2B). In the presence of active Golgi cell inhibition [61,62], GrCs tend to generate either sub-threshold responses or a limited number of spikes. In order to account for cells showing subthreshold responses, gain-frequency curves were generated for a series of parameters related to both EPSPs (average maximum depolarization, *amd*; Figure 2C) and spike properties (spike count, *sc*; spike probability, *sp*; 1st spike SD, ssd) (Figure 2C; Materials and Methods and [63]). A cumulative compound gain index (*cGI*, Figure 2D) was then generated by summing the normalized parameters for each of the recorded cells (Methods). The probability of observing a given cutoff frequency was evaluated using *amd* for EPSPs cells and *cGI* for the spiking cells. Average *amd* and *cGI* curves (Figure 2D,E) were fitted with sigmoidal functions (*cGI*: A_1_ = 1.03 ± 0.23, A_2_ = 2.69 ± 0.17, fc = 43.2 ± 13.7 Hz, *p*(χ^2^) = 0.03, *n* = 14; *amd*: A_1_ = 1.18 ± 0.05, A_2_ = 2.0 ± 0.06, fc = 77.7 ± 10.9 Hz, *p*(χ^2^) = 0.01, *n* = 24). The analysis of cells behavior showed that most of the cells displayed a cutoff frequency in the range between 20 and 50 Hz (*amd*, 45.8% out of *n* = 24 cells; cGI, 50% out of *n* = 14 cells; Figure 2F); however, all single granule cells showed high-pass filtering properties with sigmoidal responses reflecting nonlinearity in EPSP temporal summation and spike generation. Interestingly, the ensemble behavior of *amd* and *cGI* displayed a frequency dependence similar to that observed with imaging ensemble recordings [10,23], indicating that high-pass filtering network properties could be traced down to synaptic and cellular mechanisms.

The effect of long-term plasticity evoked by TBS on gain modulation was evaluated by differentially analyzing EPSPs and spiking cells. The induction of LTP tended to increase the average burst depolarization along with the probability of generating spikes and the total number of spikes emitted, which became more precise. The GrCs therefore tended to generate more action potentials at lower frequencies and with a higher precision and showed a stronger temporal summation (Figure 3A,B top traces). As a result of LTP expression, the gain profiles maintained a sigmoidal shape, with curves shifted towards higher gain values, while cutoff frequencies were anticipated for both *amd* (from 65 Hz to 47 Hz; Figure 3A bottom) and *cGI* (from 73 Hz to 44 Hz; Figure 3B bottom). Conversely, after LTD, the average burst depolarization decreased along with the probability of generating spikes and the total number of emitted spikes, which became less precise (Fig 3C,D top traces). The GrCs were therefore less prompt in generating action potentials at higher input frequencies and with a lower precision in response to a weaker EPSP temporal summation. The gain profiles maintained a sigmoidal shape while showing a shift towards lower gain values and higher cutoff frequencies in both *amd* (from 82 Hz to 179 Hz; Figure 3C bottom) and *cGI* (from 87 Hz to 95 Hz; Figure 3D bottom). On average (Figure 4), fittings through sigmoidal functions of cells undergoing LTP showed a significant decrease in the cutoff frequency (−67% cGI *n* = 8, and −50% *amd n* = 10; Figure 4A,B) and an increase in the maximum gain ( +24% *cGI n* = 8, and +16% *amd n* = 10; Figure 4A,B) and the relative gain (Gmax–Gmin)/Gmin) (41% *cGI n* = 8, and 10% *amd n* = 10; Figure 4A,B). The effect of LTP was hence to increase neuronal responsiveness by enhancing both low-frequency and high-frequency neurotransmission. Fittings through sigmoidal functions on cells undergoing LTD revealed that the cut-off frequency significantly increased (+89% *cGI n* = 5; and +82% *amd n* = 14; Figure 4D,E), while the maximum gain decreased (−13% *cGI n* = 5 and −16% *amd n* = 14; Figure 4D,E) together with the relative gain (Gmax-Gmin)/Gmin) (−13% *cGI n* = 5 and −8% *amd n* = 14; Figure 4D,E). As a result, the cumulative effect of LTD was to decrease neuronal responsiveness by reducing both low-frequency and high-frequency neurotransmission (Figure 4). The analysis of gain curves revealed that LTP increased the number of cells showing low cutoff frequencies (histogram in Figure 4C), whereas LTD increased the number of cells showing high cutoff frequencies (Figure 4F).

The average changes induced by long-term plasticity are shown in Figure 4 for both the *cGI* and the *amd*. Although, in general, the LTP-related changes appeared larger than the LTD-related changes, these results indicate that LTP and LTD can generate a bidirectional control of spike emission and filtering properties in the cerebellum granular layer.

### 3.3. Mathematical Modeling of the Impact of Plasticity on Filtering Properties

In our experimental configuration, the presence of active synaptic inhibition could lead to the induction and subsequent expression of plastic mechanisms at both inhibitory GoC-GrC [7] or excitatory mf-GoC synapses [27]. In order to further investigate the influences of long-term plasticity on gain modulation, we have employed a mathematical model of the GrC incorporating a detailed representation of the expressed ionic conductance [36], which was connected to a variable number of mfs and GoCs in a simplified version of the cerebellar circuitry (from 1 to 4 and from 0 to 4, respectively; see [36,64] and Appendix A). The input–output relationship of GrCs was evaluated by reproducing the experimental stimulation protocols (Figure 5A,B) in all the possible network configurations (see Appendix A). However, to reduce the number of combinations to be explored, the quantitative analysis was limited to four combinations of excitatory and inhibitory inputs. This choice allowed for the investigation of the whole excitatory range (one to four mf), whereas the number of GoCs was kept approximately constant (two or three). The GoC inhibition, which is known to act on GrC through a feedforward and feedback loop, expresses its actions through a polysynaptic pathway that is instantiated in a few milliseconds (3–5 ms). Furthermore, the inhibition of a single GoC is particularly intense [63], and it has been shown that the number of active GoCs poorly affects the peak of phasic inhibitory responses [36]. Moreover, according to computational estimates based on experimental in vivo recordings, the most probable number of active Golgi on a single GrC cell is between two and three (see Table 1 in [64]). The neuronal gain modulation was estimated as an I-O relationship (Figure 5C). Similar to what was observed experimentally (Figure 2B), the I-O curve of GrC responding with firing activity showed a sigmoidal shape that once fitted with a quasi-logistic function resulted in a cut-off frequency between 20 and 250 Hz (differences are due to variability in different repetitions; 95 Hz resulted from the fitting). Again, when only one mf was activated, GrCs responded with EPSPs, and the gain curve was evaluated by measuring the maximum average depolarization during spike bursts. Additionally, in this case, the average behavior was sigmoidal, with a cut-off frequency between 20 and 250 Hz (156 Hz resulted from the fitting).

The synaptic plasticity at the mf-GrC synapse is known to be expressed through a combination of presynaptic (changes in neurotransmitter release [28,42]) and postsynaptic (mainly changes in NMDA currents [28]) mechanisms. We have initially estimated the differential contribution of excitatory LTP and LTD to gain functions by alternatively increasing or decreasing the release probability of the mf-GrC synapses (Figure 6; see Methods). The main effect of synaptic plasticity on gain curves was an upward or downward shift for LTP and LTD, respectively (Figure 6, red and blue curves), which was evident in both firing and non-firing circuit configurations. Nonetheless, the changes in the cutoff frequencies displayed when simulating excitatory synaptic plasticity were not always in accordance with experimental results, especially for the non-firing circuit configuration, where the frequency actually changed in the opposite direction (from 220 Hz in the control to 260 Hz with LTP and 219 with LTD; see Figure 6A,C bottom panels for the non-firing regime; from 105 Hz to 76 Hz for LTP to 210 for LTD in the firing regime; Figure 6B,D bottom). Furthermore, despite the absolute gain changing according to the expected sign of plasticity (+21% and +55% with non-firing and firing cells with LTP and −36% and −70% with non-firing and firing cells with LTD), unexpectedly, the relative gain showed a reduction following LTP (−2% in non-firing cells and −44% in firing cells) and an increase with LTD (+21% with non-firing cells and +55% with firing cells). As a whole gain, although the qualitative evaluation of the effects of excitatory plasticity on gain transmission was in accordance with experimental observations, the quantitative evaluation of changes in gain curves suggested exploring the presence of additional mechanisms. We have thus focused our attention on plastic mechanisms in the inhibitory network.

One of the main issues preventing the experimental analysis of synaptic plasticity is the need to dissect synaptic mechanisms by pharmacologically isolating the reciprocal contribution of the different forms of plasticity on receptor targets. This limitation can be overcome by using computational models allowing for the investigation of the single contributions of ionic and synaptic currents to the emergence of changes in transmission properties. We have thus analyzed the differential contributions of pre- and postsynaptic changes that are known to be expressed following the induction of synaptic plasticity. It has been shown that, among the synaptic stages encountered by signals coming into the cerebellar cortex, different plastic mechanisms can occur. In particular, the modulation of GABAergic inhibition performed by Golgi Cells can undergo persistent changes in response to repetitive mf stimulation either by affecting mf-GoC [27] or directly affecting GoC to GrCs synapses [7]. The inhibitory plasticity at the GoC-GrC synapse has been shown to be induced postsynaptically and expressed mainly presynaptically through the retrograde diffusion of Nitric Oxide released by GrCs [7]. Since this mechanism is heterosynaptic and requires the concomitant activity of glutamatergic synapses, which typically also favor the expression of excitatory plasticity, we have decided to investigate the effect of inhibitory plasticity by changing the release probability at the GoC-GrC synapse with a sign, in agreement with changes at the mf-GrC synapse (LTP at both mf-GrC and GoC-GrC; LTD at both mf-GrC and GoC-GrC). Figure 7 shows GrCs gain curves in three mf/GoC configurations that have been adopted to account for a non-firing regime, a firing regime and a sustained firing regime. These values have been chosen according to computational estimates of the excitatory/inhibitory ratio and the most probable number of GoCs inhibiting a single GrC that has been obtained through a computational model simulating experimental data (see Table 1 in [63]). The panels of Figure 7 show that the introduction of GABAergic plasticity effectively has a homeostatic effect on GrC activity, which reverberates on gain curves. In all cases, GABAergic plasticity reduced the differences between gain curves in control conditions and during LTP or LTD (Figure 7), therefore quantitatively limiting the impact of excitatory plasticity. Interestingly, the cutoff frequencies in all cases changed according to experimental results (from 210 Hz to 190 Hz with non-firing LTP and from 210 Hz to 250 Hz with firing LTD), whereas relative and absolute gain changes were quantitatively closer to experimental changes (Figure 7).

## 4. Discussion

This work shows that long-term plasticity can tune the gain and timing transmission channels at the cerebellum input stage. Excitatory and inhibitory LTP and LTD determined how mossy fiber bursts were converted into output granule cell bursts regulating the gain and timing of output firing. These results, together with the recently shown evidence obtained with imaging and large-scale model simulations [10] regarding changes in the center-surround reorganization of signal transfer, demonstrate that the induction of long-term synaptic plasticity can effectively generate a complex *spatiotemporal* reconfiguration of granular layer transmission. This capability, which was predicted earlier only on theoretical grounds [64,65,66], did not account for the presence of long-term plasticity in the granular layer. Later, the gain control has been related to the regulation of granule cell input conductance by ionic mechanisms mainly controlled by tonic inhibition [20]. Here, we report a gain control process strictly dependent on long-term plasticity and largely extending over frequency and time domains.

### 4.1. Gain Control by Synaptic Plasticity

The granular layer transmission is tuned by LTP and LTD at different temporal and spatial levels that can be summarized as follows. In the spatial domain, the activity of neuronal ensembles is modified by long-term plasticity by a sharpening of the contrast between adjacent granular layer responding areas. The differential activity of LTP and LTD affects single-transmission lines by anticipating and increasing the response amplitude. The opposed actions of potentiation and depression, by broadening the transmission bandwidth toward low frequencies and sharpening toward high frequencies, respectively, strongly affect the time and frequency domains of single-transmission channels. Altogether, these changes reflected, at the single-cell level, the intensity of postsynaptic responses, the first-spike delay and the precision of spike emission over repeated trials. The combination of these single-cell level results with circuit-level observations, showing changes reflecting the proportion of spiking granule cells or the number of emitted spikes per cell in an active neuronal cohort [10], confirm the hypothesis that the cerebellum granular layer behaves as an *adaptable spatiotemporal filter* [35] exploiting elementary changes induced by long-term synaptic plasticity at the mossy fiber-granule cell relay.

### 4.2. Functional Implications

The essential elements of a granular layer’s computation derive from the spatiotemporal architecture of signals, which is organized mainly in sparse cores of activation with a characteristic patchy distribution of center-surround structures. According to electrophysiological and pharmacological analysis [19,23,33], these different transmission properties should reflect a variable excitatory/inhibitory ratio (E/I) that is higher in the center than in the surround because of the differential activation of glutamate and GABA receptors. This resulted in a differential voltage-dependent regulation of NMDA receptors, which can both markedly enhance burst transmission in response to their long integration time constant [21,22] and determine the induction of synaptic plasticity due to their calcium permeability [25]. In the core of activation, higher depolarization would enhance NMDA channel voltage-dependent unblock, improving synaptic integration at low frequencies and at the same time raising the calcium influx, facilitating LTP induction. In the surround, the opposite would occur, and LTD would be favored.

One of the most important aspects in neurotransmission is the modality of signal transfer, which, in the granular layer, is preferentially organized in bursts traveling in the mossy fibers and conveyed to Purkinje Cells after being spatiotemporally filtered by granule cells. Sensory stimulation activating the central cerebellar lobules tends to elicit high-frequency burst mf discharges with an average high frequency (about 100 Hz) [67]. Conversely, mfs respond with tonic, low-frequency firing to proprioceptive or vestibular activation [68]. It is therefore particularly important to understand how synaptic plasticity at the input stage of this sensory transmission line, besides tuning the frequency bandwidth, regulates the overall number of spikes emitted by GrCs and their timing. Notably, sensory inputs travelling through mfs and activating granule cells preferentially in the central lobules of the vermis have been demonstrated to operate in the millisecond time domain [69] rather than simply coding frequency-related signals. The synaptic plasticity finely tunes this operational capacity. In this scenario, it is conceivable that this finely tuned gain control can derive from a multiplicative operation implemented through short-term depression and synaptic inhibition [3]. LTP and LTD, by changing the short-term depression properties of the mf-GrC synapse, can effectively tune the transmission gain exploiting the AMPA current, and the frequency dependency could be traced back to NMDA activity [22,23]. Conversely, synaptic inhibition, by acting on spike timing and on GrC excitability through the phasic and the tonic component [70], appears to be poorly involved in gain modulation by synaptic plasticity. Nevertheless, the use of the computational model showed that changes in GABAergic inhibition mimicking plastic mechanisms were required to attenuate the marked changes in gain curves occurring when only excitatory plasticity was present (see Figure 7). As anticipated, GABAergic plasticity, by acting in a homeostatic manner, appeared to have a major effect in avoiding the saturation of synaptic weights that could occur in response to a continuous stimulation allowing for the maintenance of a dynamic range of activity in both the time and frequency domains.

The use of mathematical models to simulate neuronal and synaptic activity has been proven to be valuable in investigating conditions that are difficult to explore experimentally. Nevertheless, besides the reliability of single-cell and synaptic models that have already been demonstrated to faithfully reproduce GrCs and GoCs’ electrophysiological properties, the analysis of multi-element circuits is further complicated by the increased parameter space. One of the main limitations of such an approach is the difficulty of experimentally controlling the number of afferent excitatory and inhibitory fibers, which affect neuronal activity. We have assumed that the wide range of combinations in the mf-GrC-GoC circuit is actually limited to a restricted range (two to three mfs and two to three GoCs), leading to the simulation of just a small number of circuit architectures. We plan to further extend the functional analysis of these circuits to a more inclusive range of configurations, which could also include the pharmacological exploration of conditions that are otherwise not testable, allowing for the instantiation of a reliable benchmark for advanced translational investigations [71,72].

## 5. Conclusions

The ability of LTP and LTD to reorganize granular layer activity bears a series of functional and theoretical consequences. LTP and LTD can finetune time- and frequency-dependent properties of network transmission rather than just regulating synaptic weights, bearing relevant implications for applicative scenarios [73,74]. Our observations lend support to the hypothesis that the granular layer processes spike bursts as a relevant computational unit, in which the composition in terms of the spike delay, the number or the frequency can be finetuned. Importantly, LTP and LTD can extend the transmission bandwidth toward lower frequencies. It has yet to be understood how continuous transmission is processed. Finally, it has yet to be determined how these patterns are decoded by Purkinje cells, by molecular layer interneurons and, mostly in the vestibulo-cerebellum, by Unipolar Brush Cells.

## Figures and Tables

**Figure 1 biomedicines-10-03185-f001:**
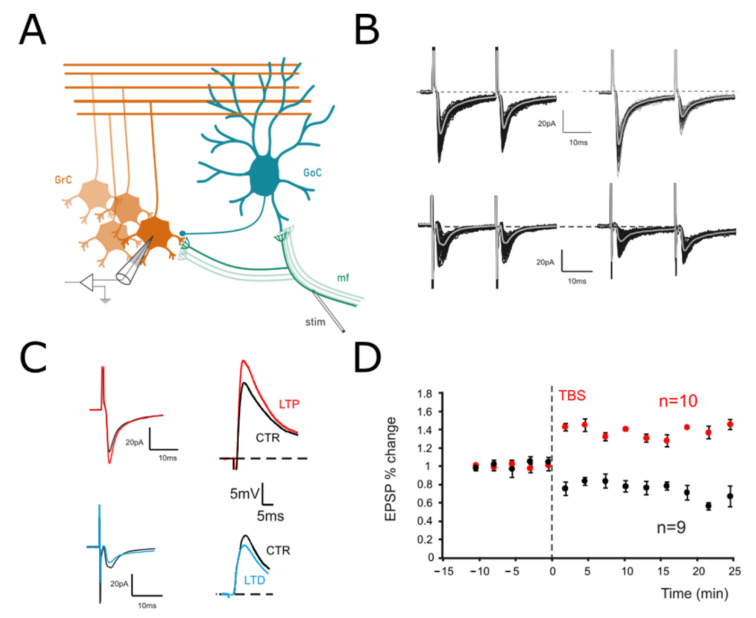
Long–Term Plasticity at the mf-GrC synapse. (**A**) Scheme of the granular layer microcircuit. The stimulating electrode (stim) is positioned onto the mossy fiber bundle (mf) in order to activate excitatory synapses. GoC, Golgi cell (local interneuron); GrC, granule cell (output cell). (**B**) *Top*. EPSCs elicited in control (left) and following TBS (right) in response to paired pulses at 50 Hz and recorded from a GrC clamped at −70 mV. In this cell, the TBS induced an increase in the average response, with a marked decrease in PPR. *Bottom*. In a different cell, EPSCs were evoked before (left) and after (right) TBS from a GrCs voltage clamped at −70 mV. In this case, TBS evoked a current depression and an increase in the PPR. (**C**) *Left*. Average responses to the first of the two stimuli shown in B are superimposed to highlight the differences between LTP (red) and LTD (blue). *Right*. EPSPs elicited by sub-threshold stimuli in the GrCs shown on the left in both the control (black traces) and after LTP (red) or LTD induction (blue). Cells were held at –60 mV (10 superimposed traces). (**D**) Time course of average EPSPs recorded from GrCs at –60 mV in response to TBS delivery. Cells undergoing LTP (*n* = 6) are represented in red, while those undergoing LTD are represented in black (*n* = 10).

**Figure 2 biomedicines-10-03185-f002:**
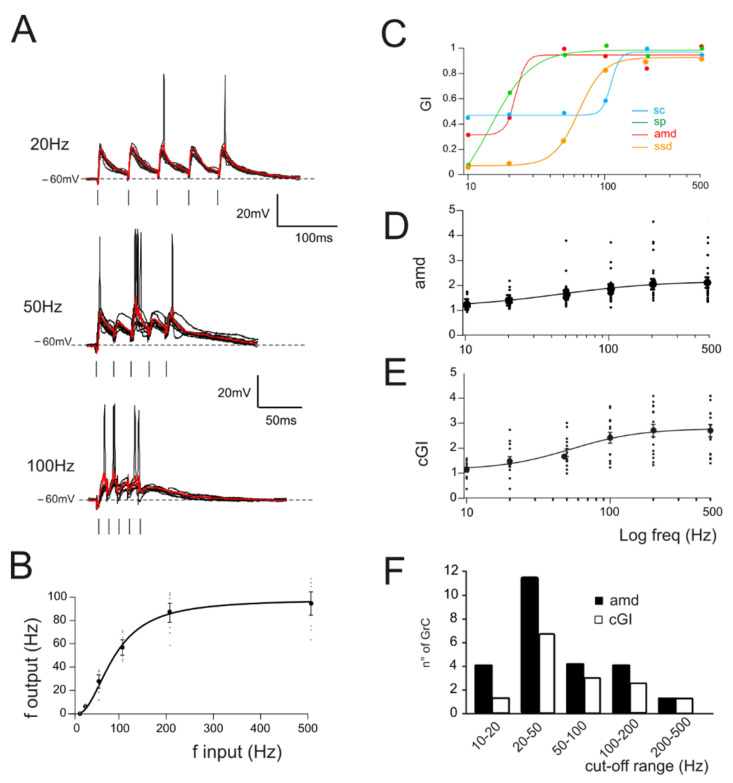
Gain curves in GrCs. (**A**) Voltage traces recorded from a single GrC stimulated with burst pulses at different frequencies. Ten traces superimposed and the average response in red. (**B**) The I/O relationship in single GrCs between the stimulation and response frequencies (*n* = 16 cells, gray circles). An average thick line is obtained with a sigmoidal fitting (Equation (1)) on average responses for each frequency (black circle). (**C**) Compound gain index (c*GI*) for the four parameters accounting for EPSPs (*amd*) and spike properties (*sc*, *sp* and *ssd*) obtained from a single GrC. (**D**) Average gain curve obtained from the analysis of subthreshold responses in all GrCs (*n* = 24). (**E**) Average gain curve of the cumulative gain indexes (*cGI*) obtained from the analysis of firing responses in all GrCs (*n* = 14). (**F**) Histogram showing the number of GrCs showing a particular cut-off frequency obtained from the fitting of *amd* (black bars) or *cGI* (white bars) curves.

**Figure 3 biomedicines-10-03185-f003:**
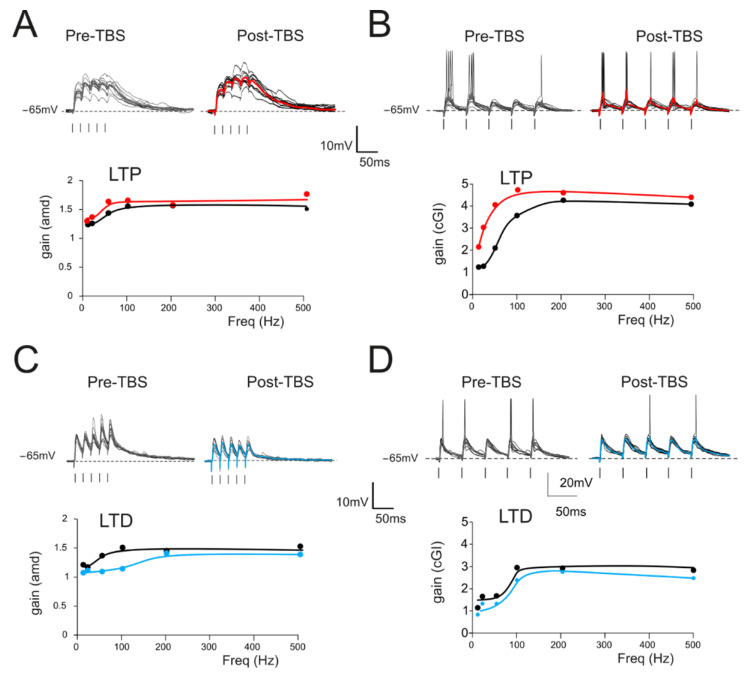
Gain modulation by synaptic plasticity (**A**) *Top*. Traces (10 repetitions) obtained from a GrC showing subthreshold responses to repetitive stimulation at 50 Hz in the control (left) and following the delivery of TBS (right). Average traces (thick, gray and red) show that, in this cell, TBS increased the average response, as expected from the expression of LTP. *Bottom.* Gain curve of the cell shown in the top traces before (black) and after LTP (red) induction. (**B**) *Top.* In a different cell, there was repetitive stimulation at 20 Hz (10 traces) obtained from a GrC showing firing responses to repetitive stimulation in the control (left) and following the delivery of TBS (right). Average traces (thick, gray and red) show that, in this cell, TBS increased the number of spikes and decreased the spike delay and variability. *Bottom.* Gain curve of the cell showed in the top traces before (black) and after LTP (red) induction. (**C**) Top. *Traces* (10 repetitions) obtained from a GrC showing subthreshold responses to repetitive stimulation at 50 Hz in the control (left) and following the delivery of TBS (right). Average traces (thick, gray and blue) show that, in this cell, TBS decreased the average response, as expected from the expression of LTD. *Bottom.* Gain curve of the cell shown in the top traces before (black) and after LTP (blue) induction. (**D**) *Top.* In a different cell, there was repetitive stimulation at 20 Hz (10 traces) obtained from a GrC showing firing responses to repetitive stimulation in the control (left) and following the delivery of TBS (right). Average traces (thick, gray and blue) show that, in this cell, TBS decreased the number of spikes and increased the spike delay and variability. *Bottom.* Gain curve of the cell shown in the top traces before (black) and after LTP (blue) induction.

**Figure 4 biomedicines-10-03185-f004:**
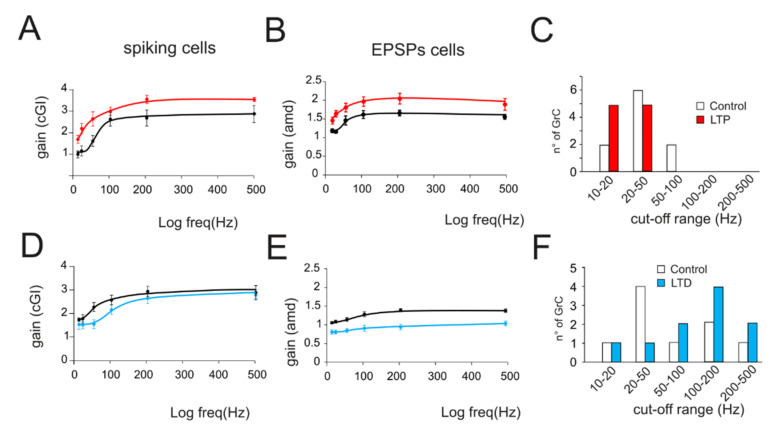
Average gain modulation. (**A**) Average change in *cGI* obtained from cells showing LTP. (**B**) Average change in the *amd* index obtained from cells showing LTP. (**C**) Histogram shows the number of cells showing a particular cutoff frequency in the control (white bars) and following the expression of LTP (red bars). Note the increase in low-frequency cells in response to LTP. (**D**) Average change in *cGI* obtained from cells showing LTD. (**E**) Average change in the *amd* index obtained from cells showing LTD. (**F**) Histogram shows the number of cells showing a particular cutoff frequency in the control (white bars) and following the expression of LTD (blue bars). Note the increase in low-frequency cells in response to LTD.

**Figure 5 biomedicines-10-03185-f005:**
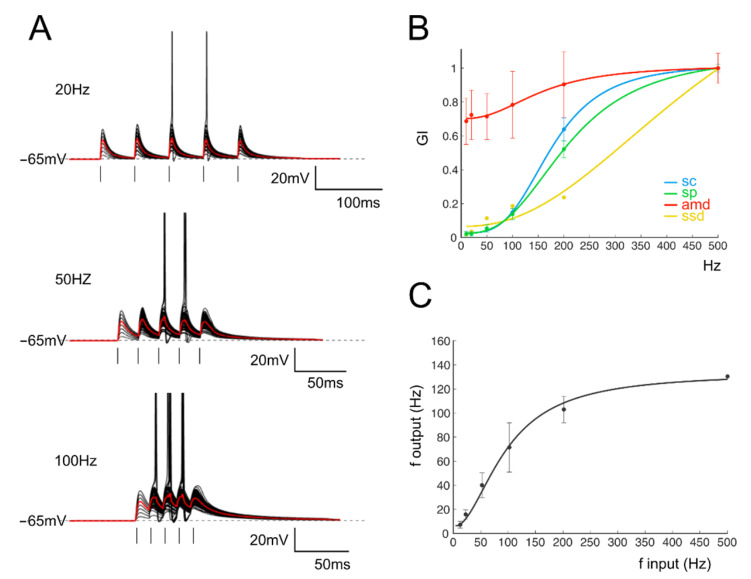
(**A**) Voltage traces show the membrane potential changes of a single GrC model stimulated with burst pulses at different frequencies. The simulated microcircuit is composed of two mfs and two GoCs. Ten traces are superimposed, and the average response is in red. (**B**) Gain index (GI) for the four parameters accounting for EPSPs (*amd*) and spike properties (*sc*, *sp* and *ssd*) obtained from the model of a single GrC. (**C**) The I/O relationship in single GrCs between the stimulation and response frequencies (*n* = 100 repetitions). An average thick line is obtained with a sigmoidal fitting (Equation (1)) on average responses for each frequency (black circle).

**Figure 6 biomedicines-10-03185-f006:**
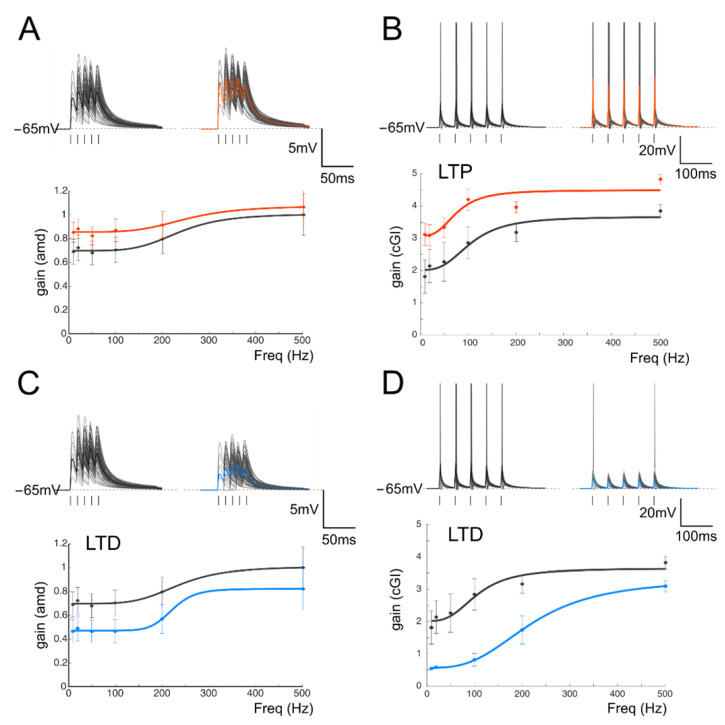
Simulation of gain modulation by synaptic plasticity. (**A**) *Top* Voltage traces obtained from a modeled GrC showing subthreshold responses to repetitive stimulation at 100 Hz in the control (left) and following LTP (right). Average traces (thick, gray and red) show an increase in the average responses. *Bottom.* Gain curves of the cell shown in the top traces before (black) and after LTP (red). The network configuration (see Appendix A) was one mf and three GoCs (**B**) *Top.* In a different network configuration (three mfs and two GoCs), repetitive stimulation at 20 Hz (10 traces) generated a prevalence of firing responses to repetitive stimulation in the control (left) and following LTP (right). Average traces (thick, gray and red) show an increase in the number of spikes and decreases in the spike delay and variability. *Bottom.* Gain curves of the cell shown in the top traces before (black) and after LTP (red). (**C**) *Top* Voltage traces obtained from a modeled GrC showing subthreshold responses to repetitive stimulation at 100 Hz in the control (left) and following LTD (right). Average traces (thick, gray and blue) show a decrease in the average responses following LTD. *Bottom.* Gain curves of the cell shown in the top traces before (black) and after LTD (blue). The network configuration (see Appendix A) was one mf and three GoCs. (**D**) *Top.* In a different network configuration (three mfs and two GoCs), repetitive stimulation at 20 Hz (10 traces) generated a prevalence of firing responses in the control (left) and following LTD (right). Average traces (thick, gray and blue) show a decrease in the number of spikes and increases in the spike delay and variability. *Bottom.* Gain curves of the cell shown in the top traces before (black) and after LTD (blue).

**Figure 7 biomedicines-10-03185-f007:**
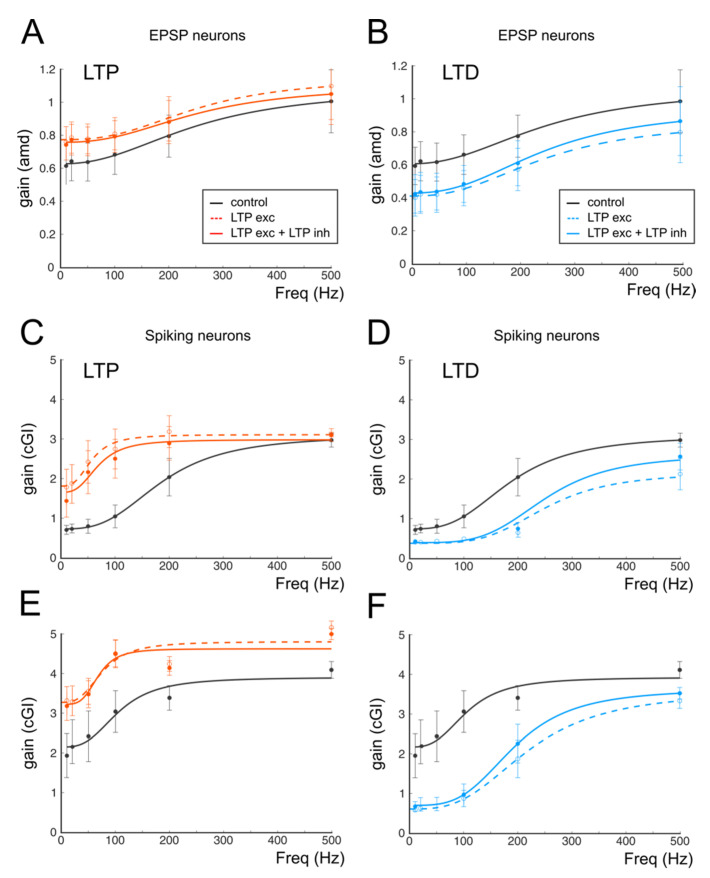
The effect of GABAergic plasticity on gain curves. (**A**) Gain curves of a cell receiving a combination of one mf and three GoCs and responding with EPSPs. Black trace (control); red dashed line (excitatory LTP); continuous red line (excitatory and inhibitory LTP). (**B**) In the same cell shown in A (one mf and three GoC), purely excitatory LTD (dashed blue line) induces a marked shift to lower gain values, which is buffered by the concomitant inhibitory LTD (continuous blue line). (**C**) Gain curves of a cell receiving a combination of two mfs and two GoCs and responding with spikes. Black trace (control); red dashed line (excitatory LTP); continuous red line (excitatory and inhibitory LTP). (**D**) In the same cell shown in C (two mfs and two GoCs) purely excitatory LTD (dashed blue line) induces a marked shift to lower gain values, which is buffered by the concomitant inhibitory LTD (continuous blue line). (**E**) Gain curves of a cell receiving a combination of three mfs and two GoCs and responding with EPSPs. Black trace (control); red dashed line (excitatory LTP); continuous red line (excitatory and inhibitory LTP). (**F**) In the same cell shown in C (three mfs and two GoCs) purely excitatory LTD (dashed blue line) induces a marked shift to lower gain values, which is buffered by the concomitant inhibitory LTD (continuous blue line).

**Table 1 biomedicines-10-03185-t001:** GrC voltage-dependent conductance.

Ionic Channel	Conductance, nS	Localization
Na	8.589	Axon
K_DR_ K-A K-IR K-Ca Ca K-slow Lkg1 Lkg2	2.338 2.336 3.382 0.951 0.707 0.108 0.264 0.17 0.042	Hillock Axon, Hillock Soma Soma Dend(4) Dend(4) Soma All compartments Dend(1,2,3,4)

## Data Availability

Data can be made available upon direct request to the corresponding author.

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
