# Peer review of "Long-Term Synaptic Plasticity Tunes the Gain of Information Channels through the Cerebellum Granular Layer"

_biomedicines, 2022, doi:10.3390/biomedicines10123185_

Round 1

Reviewer 1 Report

In the manuscript submitted, Mapelli and Colleagues study the impact of long-term synaptic plasticity at the input stage of cerebellum and aim at demonstrating that LTD and LTP can effectively shape the dynamics of transmission in granule cells at the single cell scale. By combining electrophysiological patch-clamp recordings and modelling, they show that LTP and LTD can affect the gain and timing of granule cells transmission in response to mossy fibers stimulation. The authors present the experimental results, describing the effect of stimulation of mossy fibers inputs while recording the electrophysiological responses in granule cells with the patch-clamp technique. They apply a protocol for LTP or LTD induction and they analyze their impact on the granule cell response. They found that long-term plasticity affects action potential latency, precision and spike probability with the result of tuning timing and frequency properties of granule cells output. In the second part of the manuscript, a mathematical modelling, which includes granule cells and Golgi cells contributions, based on current and previous findings, is used to model the granule cells output.  Granule cell activity is simulated under different configurations by varying the number of mossy fibers and Golgi cells inputs.

The scientific advances presented in this paper are important to move further in understanding the processing performed at the mossy fibers-granule cells stage in the cerebellum. The results and the modeling address some previously uncovered aspects of the role of ltd and ltp in granule cells processing, by also including in the model the plasticity in inhibitory Golgi network. The experimental recordings are of excellent quality and the modelling reproduces the data quite well in general. The manuscript in all the sections is very well written, clear and results are well presented. I recommend the manuscript for publication after some minor revisions.

Here I list some specific comments:

-     -  Mossy fibers can fire with short high-frequency bursts or by frequency-modulated discharges, depending on the nature of the stimulus and their origin. It would be useful for the readers to know the lobules where recordings have been performed and the mossy fibers firing frequencies observed in that specific area by citing previous works if available.

-     - Figure 1A: MF stimulation in this experimental condition has a high chance to activate Golgi cells too. To have a better representation of the experimental configuration, authors should correct the scheme and include more than one MF stimulation and the activation of the MF branch on the Golgi cell represented.

-          Lines 224-226: can authors provide the data or a refence for the time constant of recovery?

-          Lines 249-251: how were the values obtained?  

-          Lines 433-435: the decision to partially explore the GoCs range because it is not affecting substantially the phasic component is an important point, that authors may consider to strengthen by adding a reference or supporting data or a rational in the text.

-          Figure 7: Authors should explain in the Results section the motivations for retaining these specific configurations of MF and GoC number of inputs (1mf 3goc, 2mf 2goc, 3mf 2gc).

-          Some typos need to be corrected in text and figure 1 legend.

The abbreviation “ns” is used in the Methods section, but in the main text it is quoted as “sc”. Correct reference 45: authors missing.

Figure 5A and B are not cited in the text.

Reviewer 2 Report

The present article combined the use of experimental and theoretical strategies in order to understand how LTP and LTD may influence the gain of nervous communication. The hypotheses are well explored experimentally. The only point that the authors should explain in detail refers to the mathematical model used in this work. Therefore, it is necessary to present even briefly the mathematical model reported with the gain changes as described in the model proposed in [47]. Even in the reference [47] itself does not provide a detailed or even reasonable quantitative description, but only a brief description of the model. This is not sufficient!! Since this is a fundamental part of the work, authors must include a detailed quantitative explanation of all models assumed in the investigations. This improvement will allow a rigorous comparison between the experimental results and the models, also contributing to a better comprehension of the phenomenon and findings presented in this report.

Minor considerations: You have to say the meaning of TBS in 2.2 and not on page 6.
